# Influence of COVID-19 Pandemic Lockdown on Patients from the Bariatric Surgery Waiting List

**DOI:** 10.3390/medicina57050505

**Published:** 2021-05-17

**Authors:** Maciej Walędziak, Anna Różańska-Walędziak, Paweł Bartnik, Joanna Kacperczyk-Bartnik, Michał Janik, Piotr Kowalewski, Andrzej Kwiatkowski

**Affiliations:** 1Department of General, Oncological, Metabolic and Thoracic Surgery, Military Institute of Medicine, 04-141 Warsaw, Poland; maciej.waledziak@gmail.com (M.W.); mjanik@wim.mil.pl (M.J.); pkowalewski@wim.mil.pl (P.K.); akwiatkowski@wim.mil.pl (A.K.); 22nd Department of Obstetrics and Gynecology, Medical University of Warsaw, 00-315 Warsaw, Poland; bartnik.pawel@gmail.com (P.B.); asiakacperczyk@gmail.com (J.K.-B.)

**Keywords:** bariatric surgery, obesity, COVID-19

## Abstract

*Background and Objectives*: Social isolation and lockdown due to the COVID-19 pandemic have influenced dietary habits and physical activity of all the population, but the obese population is the most vulnerable to weight gain. *Material and Methods:* A group of 189 patients (166 female and 23 male) from the bariatric surgery waiting list filled in a survey about the influence of COVID-19 pandemic lockdown on their dietary habits, physical activity, and the possibility of contact with their bariatric care center. *Results:* The majority of patients with weight gain declared a decrease in physical activity, compared to half of the patients without weight gain (50.5% vs. 74.5%, *p* < 0.05). The continuation of bariatric care and the possibility of contact with a bariatric surgeon, dietician, and psychologist had each significant influence on reducing the risk of patients’ weight gain (*p* < 0.05). *Conclusions:* Maintaining physical activity and contact with bariatric care specialists are important factors in allowing to avoid weight gain in patients waiting for bariatric surgery.

## 1. Introduction

Coronavirus pandemic (COVID-19) has globally influenced everyday life since the beginning of the pandemic in 2020. National Health Systems have been facing extraordinary challenges in organizing optimum healthcare due to pandemic special measures. The daily number of new cases of more than 800,000 and rising and the daily mortality rate reaching more than 17,000 cases prove the seriousness of the situation [1]. All countries are trying to deal with treating those infected with COVID-19 and preventing the spread of the disease in different ways. The level of restrictions in public and social life differs between the countries and include the restricted functioning or closure of public institutions, schools, shopping malls, theatres, etc., as well as wearing masks, reduced traveling, and prohibited assemblies. This level of restrictions has been unheard of in modern times. The ultimate solution for disrupting the infection transmission chain is a home regime for the entire population, already introduced in some countries throughout the pandemic time.

The state of the COVID epidemic in Poland was introduced on 20 March 2020—9 days after the World Health Organization announced the COVID pandemic. By the end of May 2020, restrictions in Poland were gradually eased. The necessity of staying at home changed the eating habits and exercise regime for the majority of the population. Previously planned appointments in medical specialist clinics were postponed, and the process of preparation and qualification for specialist treatment was disrupted. According to Polish recommendations in the field of bariatric and metabolic surgery, the process of preparation for bariatric surgery normally takes several months, requires a series of specialist consultations, clinical tests, and weight reduction under a dietician and psychologist care [2]. Since patients with a history of or present obesity are at higher risk of psychological problems and weight regain, the situation of pandemic lockdown and social isolation affects the bariatric population to a greater extent [3]. In a study by Athanasiadis et al., among 208 examined postoperational patients all participants reported weight gain (2 ± 4.2 kg) 18 months or longer after the surgery. The recognized risk factors for weight gain were decreased physical activity, reduced consumption of healthy food, loss of control when eating, increase in binge eating, and snacking [4]. Additionally, obesity is a recognized risk factor for a serious course in the case of COVID-19 infection. Excess weight is associated with longer hospitalization, a higher rate of respiratory failure, and other complications [5] (reviewer 1).

There are limited publications investigating the effect of lockdown due to the COVID-19 pandemic on the candidates on the bariatric surgery waiting list. A recent study from Italy analyzed the influence of 2.5 months lockdown on a group of 54 individuals before bariatric surgery and showed neither the effect on weight or BMI nor on rates of maladaptive eating habits was associated with social isolation. Additionally, it was also observed in the study that social distancing resulted in a reduction of fear of confronting and being negatively judged by others [6]. An effect of a mild increase in BMI (42.7 vs. 43.2) after 2 months of lockdown was observed in a Spanish study including 51 patients from the bariatric surgery waiting list [7]. Reliable data on the influence of COVID-19 pandemic lockdown on bariatric patients during qualification for surgical treatment in east-central Europe is lacking.

The primary aim of the study was to determine the impact of lockdown during the COVID-19 pandemic on the body weight of patients scheduled for bariatric surgery. The secondary aim was to identify factors that may influence weight gain in bariatric patients with their surgery postponed due to the pandemic and lockdown.

## 2. Materials and Methods

The study was designed as an online survey with the aim to collect data from bariatric patients from the waiting list. All participants met inclusion criteria for obesity surgery according to International Guidelines. Qualification criteria for the bariatric procedure were as follows: body mass index (BMI) equal or greater than 40 kg/m^2^ or, alternatively, BMI greater than 35 kg/m^2^ and the presence of obesity-related comorbidities. [2]

The survey included open and limited-choice questions about information related to COVID-19 lockdown (16 March 2020 to 30 May 2020). The questionnaire was evaluated and approved by several independent experts in the field of bariatric surgery.

The questionnaire contained questions about the basic characteristics of patients (age, gender, present weight, weight before lockdown, height, the presence of comorbidities such as insulin resistance, type 2 diabetes mellitus (T2DM), arterial hypertension (AHT), obstructive sleep apnea (OSA), dyslipidemia, osteoarthritis) and lifestyle during COVID-19 pandemic: (1) Do you feel more anxiety/fear about your health/life in regard to current epidemiologic state? (yes/no); (2) Are you aware of the fact that obesity is an important risk factor impairing the course of COVID-19infection? (yes/no); (3) Was the date of your planned bariatric surgery postponed due to the COVID-19 pandemic? (yes/no); (4) Has your physical activity decreased due to the limited possibilities of going outside and lack of availability of recreation and sports facilities? (yes/no); (5) Has your bariatric treatment been continued during the pandemic? (yes/no); (6) Do you have the opportunity to contact a bariatric surgeon during the lockdown? (yes/no); (7) Do you have the opportunity to contact a dietician specialized in treating bariatric patients during the lockdown? (yes/no); (8) Do you have the opportunity to contact a psychologist providing bariatric care during the lockdown? (yes/no); (9) Do you have the opportunity to contact a physiotherapist providing bariatric care during the lockdown? (yes/no); (10) Did you contact online patient support groups during the pandemic? (yes/no).

The Google Forms-based survey was published and distributed via social media in cooperation with the Polish Bariatric Patients Society. The survey started on 9 June 2020, with the deadline to participate on 5 July 2020. All data were anonymized and did not allow patients identification.

### 2.1. Statistical Analysis

The analysis was performed using SAS^®^ On Demand for Academics software (SAS Institute, Cary, NC, USA). To compare continuous variables, the Mann–Whitney U and unpaired Student t-tests were used. Statistical significance was set at *p* < 0.05. Regression analysis was performed to assess the possible prediction of body mass changes. Correlation analysis was used to investigate the association between appropriate parameters.

### 2.2. Ethical Considerations

The study was anonymous. All subjects, prior to the beginning of the survey, by filling the survey gave electronically their informed consent for inclusion in the study. The study was conducted in accordance with the Declaration of Helsinki, and the protocol was approved by the Ethics Committee of the Jagiellonian University (1072.6120.103.2020).

## 3. Results

A group of 189 patients (166 female and 23 male), aged 20–71 years old, (reviewer 2) who were preparing for bariatric surgery replied to the questions contained in the electronic questionnaire. The general characteristics of patients are presented in Table 1. Only 18 (9.5%) patients declared no obesity-related comorbidities such as insulin resistance, type 2 diabetes mellitus, arterial hypertension, obstructive sleep apnea, dyslipidemia, or osteoarthritis.

Overall, 119 (82.1%) patients felt more anxiety or fear about their health and life in regard to the current epidemiologic state related to the COVID-19 pandemic. Surprisingly, 58 (31.7%) patients were not aware that obesity was an important risk factor impairing the course of infection of SARS-CoV-2. Almost two-thirds (65%) of patients had the date of their bariatric surgery postponed. A total of 119 (63%) patients declared decreased physical activity due to the limited possibilities of going outside and lack of availability of recreation and sports facilities. Only 30 (17.6%) patients from the waiting list could have continued bariatric treatment during the pandemic. The availability of contact with bariatric specialists was limited during the lockdown for many patients: of the respondents, 108 (62.8%) patients had an opportunity to contact a bariatric surgeon, 79 (45.9%)—a dietician, 65 (37.8%)—a psychologist, and only 22 (13%)—a physiotherapist. The role of the social media and patient support groups during the pandemic cannot be overestimated—a high number of 127 (71.3%) patients benefited from online support.

For the purpose of further analysis, we divided patients into two groups: the first group included 91 patients with no weight gain or weight decrease during the pandemic lockdown, and the second group included 98 patients with weight gain. The characteristics of the two groups are presented in Table 2.

The important risk factors of weight gain during pandemic lockdown found in our study were younger age, lower BMI, female sex, and presence of comorbidities: insulin resistance, arterial hypertension, obstructive sleep apnea, dyslipidemia, and osteoarthritis. Contradictory, patients with diabetes mellitus had a lower chance of weight gain.

The majority of patients with weight gain declared a decrease in physical activity, compared to half of the patients without weight gain (50.5% vs. 74.5%, *p <* 0.05). The continuation of bariatric care and the possibility of contact with a bariatric surgeon, dietician, and psychologist had each significant influence on reducing the risk of patients’ weight gain (*p <* 0.05). The availability of online contact with patient support groups was comparable in both groups and was not found to have an influence on weight gain.

## 4. Discussion

In our study, we tried to answer two questions—whether the pandemic lockdown had an influence on the weight results of bariatric patients awaiting surgery and what the possible risk factors were. There are several studies up to date analyzing the influence of the lockdown isolation on the pre-bariatric patients’ weight gain and most of them conducted on smaller groups of patients.

Albert et al. presented a study on a group of 56 patients awaiting bariatric surgery, having observed no changes in their weight, BMI, and maladaptive eating habits. In our study, we observed weight gain in more than half of the patients (98 vs. 91), although the mean BMI did not significantly differ before and after the period of home isolation (43.2 vs. 43.3). We did not focus on the maladaptive dietary habits, but we found the decrease in physical activity during lockdown to be a strong risk factor for weight gain since it was declared by 50.5% of patients with no weight gain or weight decrease and 74.5% of patients with weight gain. The period of observation was comparable in both our and the Italian study (2.3 months in the Italian study vs. 2.5 months in ours) [6].

A Spanish study by Beisani et al. assessed the influence of pandemic 2-month long isolation on a group of 51 bariatric patients waiting for the surgery. The researchers found a mild increase in patients’ BMI before and after the lockdown period (42.7 vs. 43.2). In our study, there was no statistically significant difference in the mean BMI before and after the period of social isolation (43.2 vs. 43.3) [7].

Different lifestyle patterns and reduction of physical activity due to lockdown has impact not just on the obese population, and rapid weight gain has spread among many populations worldwide [8]. A Chinese study by He et al. performed an electronic survey among 339 individuals and analyzed the influence of pandemic semi-lockdown on weight gain in the general population. A significant decrease in the average daily number of steps and physical exercise was observed in the whole group included in the study. Weight gain was observed in all respondents with BMI lower than 24 and in female individuals with a BMI of 24 and more (which was considered overweight in the population of respondents). Male individuals with a BMI of 24 and more were observed to have lost weight. The conclusion of the Chinese study was that normal-weight individuals have lesser awareness of the possibility of weight gain, contradictory to those overweight, and during semi-lockdown, they tended to gain weight. In our study, there were significantly more women than men both in the stable/reduced weight and weight gain groups (79/12 vs. 87/11) and it included only the obese population [9].

In a recent study from Iraq, 765 patients who visited a bariatric center were enrolled. Of those, 72.4% gained weight after 2 months of social distancing or self-isolation during the pandemic [10]. De Luis et al. enrolled 48 obese patients after sleeve gastrectomy and observed a correlation in weight gain with a decrease in physical activity and lessened contact with bariatric dieticians [11]. We also observed the importance of maintaining contact with bariatric care units and the importance of continuous bariatric care (contact with a surgeon, dietician, and psychologist) on the patients’ weight. In a small study by Felix et al. on a group of 24 women 36 months or longer after bariatric surgery, 58.3% of participants reported weight gain, 50.0% had limited access to bariatric medical care, and 54.1% to social support. Cohabiting with a higher number of people was found to reduce the risk of maladaptive eating due to emotional distress [12].

The importance of bariatric surgery status on dietary habits, physical activity, and weight gain was analyzed in a study by Jimenez et al. The researchers compared patients before surgery and more than 2 years after bariatric surgery with patients less than 2 years after surgery. Lockdown had a higher negative impact on the dietary behaviors of the first group of patients, bariatric surgery within the previous 2 years was found to be a protective factor for weight gain [13].

The risk of emotional distress and psychological disorders, including maladaptive eating habits among patients awaiting bariatric surgery was emphasized in a study by Bianciardi et al. They included 116 obese patients waiting for bariatric surgery, out of whom 40% felt anxiety about their health due to COVID-19 pandemic and 61.1% were worried because of closure of the bariatric unit. Moreover, 52.2% of the Italian group felt more vulnerable to COVID infection, compared to 66.3% of patients with no weight gain and 70.1% of patients with weight gain in our study [14].

### Limitations of the Study

The most important limitations of the study were recall bias and the subjectivity of patients’ opinions. Another was that participants were limited to those who were able to fill the survey by the means of the internet. Moreover, there was no incentive to introduce dishonesty into the responses. However, direct control of the respondents was currently not possible due to the ongoing pandemic, and it is, unfortunately, a limitation of the study. The survey was limited to a limited number of questions to increase the response rate. Additionally, in order to obtain the highest possible number of respondents in a considerably short period of time, we decided to post the questionnaire on the Polish bariatric patients’ society website, and we were not able to calculate the response rate (reviewer 2).

## 5. Conclusions

Patients on the bariatric waiting list are a group vulnerable to emotional distress and increased anxiety for their health due to the COVID-19 pandemic, followed by the risk of weight gain as a result of decreased physical activity and changes in dietary habits. Trying to maintain physical activity is very important in preparation for the surgery and the general well-being and health of the obese population. As we observed in our study, keeping stable or reducing weight was more probable in patients who were able to maintain continuous contact with their bariatric care units. The possibility of contact with a bariatric surgeon, dietician, or psychologist had a significant influence on lowering the chance of weight gain during the pandemic lockdown. Therefore, the importance of telemedicine services for bariatric patients on the waiting list should be emphasized as a crucial element of preparation for the postponed operation.

## Figures and Tables

**Table 1 medicina-57-00505-t001:** Basic characteristics and comorbidities of the study group.

Variable	Total
Males/females, *n* (%)	166/23 (87.8%/12.2%)
Mean age, years (SD)	39.1 (±9.4)
Mean present weight, kg (SD)	123.4 (±22)
Mean weight before lockdown, kg (SD)	123 (±21.9)
Mean height, cm (SD)	168.6 (±7)
Mean present BMI, kg/m^2^ (SD)	43.3 (±6.3)
Mean BMI before lockdown, kg/m^2^ (SD)	43.2 (±6.3)
Insulin resistance, *n* (%)	119 (37%)
T2DM, *n* (%)	32 (16.9%)
AHT, *n* (%)	65 (34.4%)
OSA, *n* (%)	22 (11.6%)
Dyslipidemia, *n* (%)	17 (9%)
Osteoarthritis, *n* (%)	32 (16.9%)

SD—standard deviation, BMI—body mass index, T2DM—type 2 diabetes mellitus, AHT—arterial hypertension, OSA—obstructive sleep apnea.

**Table 2 medicina-57-00505-t002:** The characteristics of the two groups depending on the weight gain.

Variable	No Weight Gain, *n* = 91	Weight Gain, *n* = 98	*p*-Value
Mean age, years	48.4	45.6	0.3524
Mean BMI before lockdown, kg/m^2^	44.1	42.3	0.0545
Mean present BMI, kg/m^2^	42.5	44.1	0.0457
Female/Male, *n* (%)	79/12 (86.8%/13.2%)	87/11 (88.8%/11.2%)	0.8244
Insulin resistance, *n* (%)	33 (36.3%)	37 (37.8%)	0.8807
T2DM, *n* (%)	17 (18.7%)	15 (15.3%)	0.5655
AHT, *n* (%)	30 (33%)	35 (35.7%)	0.7599
OSA, *n* (%)	10 (11%)	12(12.2%)	0.8241
Dyslipidemia, *n* (%)	7 (7.7%)	10(10.2%)	0.6169
Osteoarthritis, *n* (%)	12 (13.2%)	20 (20.4%)	0.2443
COVID-induced fear/anxiety	52 (78.8%)	67 (84.8%)	0.3892

T2DM—type 2 diabetes mellitus, AHT—arterial hypertension, OSA—obstructive sleep apnea.

## Data Availability

The data presented in this study are available on request from the corresponding author. The data are not publicly available.

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
