# Peer review of "Influence of COVID-19 Pandemic Lockdown on Patients from the Bariatric Surgery Waiting List"

_medicina, 2021, doi:10.3390/medicina57050505_

Round 1

Reviewer 1 Report

Interesting manuscript.

This is a description of a survey involving  189 patients from the bariatric surgery waiting list.

The loss of specialist contact clearly shows that conduct to a weight gain for these predisposed patients.

Finally the study claims for further methods to keep the patients compliant with the prescribed therapies and suggested habits.

I would suggest very few changes.

" Additionally, 54
obesity is a recognized risk factor for serious illness in case of COVID-19 infection, hos- 55
pitalization, respiratory failure and other complications"

i would say manifestation or syndrome more than illness. than covid is already a syndrome not infection. i would make more precise the definitions also elsewhere.

Not really big changes to suggest

Author Response

Dear reviewer 1,

Thank you for your remark. We changed manuscript as suggested.

Reviewer 2 Report

In this manuscript, the authors reported the impact of lockdown during the COVID-19 pandemic on body weight and the factors that influence the weight gain of patients scheduled for bariatric surgery. The presented work is interesting but needs minor revision. Comments and critiques below are intended to help the authors to improve their study.

  1. Did the author know the employment status of the participants? In my view, employment status also affects emotional and anxiety behaviors. Also, did the author include their educational background in their questionnaires?
  2. Can authors also provide any information related to the medication they are taking for weight loss?
  3. “Categorical variables were compared using the χ2 and Fisher's exact tests”, however; authors did not talk about them in their results and discussion. Please include them in the manuscript.
  4. Page 4, Lines 148, 151: The p-value should be italicized in the entire manuscript.
  5. Authors should provide the participant's age in range, such as X-Y years.
  6. Authors need to include any limitations found in the designed study.

Author Response

Dear rewiever 2,

Thank you very much for your remarks. We chanced the manuscript accordingly.

  1. Unfortunately, we did not collect data about the employment status of participants. The survey was limited to only a few questions to increase the response rate.
  2. The participants did not take any drugs having effect on weight loss, the routine supplementation after bariatric surgery was limited to vitamins, proteins and micronutrients.
  3. We changed "Statistical Analysis" paragraph
  4. We changed manuscript according to suggestion
  5. We added the age range of the participants.
  6. We add "Limitations of the study" paragraph as you suggested.
